# Evaluation of Well Improvement and Water Quality Change before and after Air Surging in Bedrock Aquifers

Kyoochul Ha [1,2], Hyowon An [1,2], Eunhee Lee [1], Sujeong Lee [1,2], Hyoung Chan Kim [1] and Kyung-Seok Ko [1,2,*]

1  Groundwater Environment Research Center, Korea Institute of Geoscience and Mineral Resources, Daejeon 34132, Korea; hasife@kigam.re.kr (K.H.); ahw@kigam.re.kr (H.A.); eunheelee@kigam.re.kr (E.L.); crystal2@kigam.re.kr (S.L.); khc@kigam.re.kr (H.C.K.)

2  Department of Mineral & Groundwater Resources, University of Science and Technology, Daejeon 34113, Korea

*  Correspondence: kyungsok@kigam.re.kr; Tel.: +82-42-868-3162

**Abstract:** When a drought occurs, drought response is mainly focused on the development of new wells. However, it is inefficient to respond to droughts by developing additional new wells in areas where many existing groundwater wells have been developed. Rather, it is necessary to find a way to use the existing wells efficiently and, if possible, increase the amount of groundwater that can be pumped. In this study, a pumping test and analysis method were used to evaluate the effect of air surging on improving existing wells. Drawdowns were reduced in the test wells, and, accordingly, the average specific discharges and transmissivities were increased. Since many factors in bedrock aquifers must be considered in order to calculate the well efficiency for the evaluation of the well performance, it seems better to compare the pumping rate and drawdown based on a reference time calculated by an adjusted time. Such factors could be the uncertainty of the aquifer model, aquifer inhomogeneity, and a hydrogeologic boundary. Additionally, in this process, the changes in groundwater quality were investigated, as well as the substances that caused the degradation of the well performance in bedrock aquifers. According to the results of the groundwater quality analysis conducted during the surging process and the step drawdown tests, there was no significant groundwater quality change before and after surging, but it appeared that there was an inflow of contaminants from the upper shallow strata close to the surface. According to the results of the XRD, XRF, and SEM-EDS analyses for the substances collected during surging and the substances deposited inside the well pipe, most of the substances were Fe-related amorphous components. Additionally, Fe coexisted with components such as As, V, and Zn, which formed the well casing together with Fe and were eluted in the surging process and step drawdown tests.

**Keywords:** well improvement; air surging; well rehabilitation; step drawdown test; bedrock aquifer



## 1. Introduction

In the event of a drought in Korea, drought response mainly focuses on the development of new wells. However, it is inefficient to respond to droughts by developing additional new wells in areas where many existing groundwater wells have been developed. Rather, it is necessary to find a way to use the existing wells efficiently and, if possible, it is better to increase the amount of groundwater that can be pumped. In addition, it is worth considering a plan to utilize the existing wells through water quality evaluation using the recently developed data-driven technique to determine whether the appropriate water quality is satisfied for each well [1]. In this study, the pumping test and analysis method were considered for evaluating the effect of air surging to improve the existing wells. Additionally, in this process, changes in groundwater quality were investigated, as well as the substances that caused the degradation of the well performance in bedrock aquifers.

When a well is developed and used for a long time, the casing inside the well becomes corroded or various substances become deposited on the screen, causing clogging that degrades well performance. Well clogging is caused by decreasing amounts of water pumped in the same drawdown in the well, and it also increases the hydraulic gradient and flow velocity around the well [2,3]. Subsequently, the specific yield of the well (well efficiency), which is defined as the ratio of the pumped quantity to the drawdown, decreases [4]. Such a phenomenon is commonly known as well aging. Well aging is caused by a variety of physical, chemical, and biochemical processes [5,6]. These processes include encrustation from mineral deposits; biofouling caused by microbial growth; and the physical clogging of nearby wells due to sediments, well or casing corrosion, carbonate and aluminum hydroxide deposits, and/or iron and manganese deposits [7–9].

Well rehabilitation is defined as the measures taken to correct clogging problems in a well (restoration or regeneration) [5]. Generally, there are two main categories of well rehabilitation: chemical and physical (mechanical). In chemical rehabilitation, the encrusting material is dissolved using inorganic or organic acid mixtures, which are pumped into the well and left until the coatings are dissolved. Chemical rehabilitation has the major disadvantage that most chemicals are harmful to the environment. Physical methods include attaching a brush to a drill with high-pressure jetting, hydrofracturing, and surging [10]. In recent years, one of the technologies categorized as a physical method, the ultrasonic method, has begun to be used for well rehabilitation [11].

In Korea, the use of groundwater for agricultural purposes is concentrated during a specific period of rice farming in summer, rather than continuously throughout the year. Because of this irregular usage pattern, it is necessary to periodically check whether the well can be operated normally and to take some measures to properly maintain and manage the well conditions. Therefore, according to the National Groundwater Act, agricultural wells with a pumping rate exceeding 150 $m^3$/day are stipulated to be inspected and maintained through follow-up management every 5 years. As a well rehabilitation method, air surging is generally used. In order to quantitatively evaluate well performance or well yield, the step drawdown test is usually conducted [6,12].

Although many technological developments for well rehabilitation are currently being made worldwide, there are still issues to be resolved. In particular, hydrogeologic characteristics should be considered to identify what causes the deterioration of the pumping wells developed in the bedrock aquifer, and an appropriate method to evaluate the improvement by well rehabilitation should be developed. In this study, as an extension of this topic, when air surging, which is most commonly used as a well rehabilitation method, is applied to a pumping well developed in a bedrock aquifer, methods to evaluate its effect for improving the well are presented. In this process, geochemical and microscopic studies were conducted on substances that cause the aging or clogging of wells. This study intends to present a comprehensive assessment of well performance or well yield, changes in water quality before and after air surging, and an analysis of clogging substances. The improvement effects of air surging are mainly achieved by comparing the well efficiencies by the step drawdown tests. However, in bedrock aquifers, it is difficult to estimate well efficiency because the aquifers are not homogeneous, and the hydrogeologic boundary conditions are varied. Therefore, this study presented a practical evaluation methods for step drawdown test. In addition, we intend to contribute to future well maintenance by providing information on substances that degrade well performance.

## 2. Materials and Methods

### 2.1. Study Area

The study area is about 4 km away from the coast in the mid-western region of the Korean Peninsula, which belongs to Hongseong County and has an area of 2.83 $km^2$ (283.3 ha). The area consists of a hilly mountainous area with an elevation of less than 100 m above sea level and relatively flat farmland. In terms of land use, forest areas account for 189.1 ha (66.7%); rice paddy fields and upland farming areas account for 49.5 ha (17.5%) and

37.7 ha (13.3%), respectively; and other residential areas, roads, and water streams account for 6.94 ha (2.4%). As a typical rural village in Korea, paddy farming and upland farming are carried out in flat areas surrounded by mountains. Sesame, red pepper, and sorghum are the main upland crops, and the types of the crops have diversified in recent years [13,14]. The geology of the study area is composed of the Paleozoic Devonian Taean Formation, Mesozoic Triassic granite and syenite, Late Triassic to Early Jurassic Metamorphic rocks, Jurassic two-mica granite, Jurassic volcanic tuff, and Quaternary alluvium that cover the strata with unconformities. In addition, three faults are distributed in this area, and the degree of fracture development related to these faults can influence the characteristics of the groundwater abundance of the bedrock aquifer in the study area [15].

There are a total of 107 developed wells in the area: 71 are for agricultural irrigation, of which 31 are for rice farming, 28 are for other upland crops, and 12 are for other uses. There are 42 pumping wells mainly used in the area, as shown in Figure 1, and the admitted groundwater usage is 418 m$^3$/day for living and 2178 m$^3$/day for agriculture. However, the actual groundwater pumpage is less than 20% per year, and since rice farming is the main activity, a high amount of irrigation water is required for paddy farming in summer. When it does not rain during this time, there will inevitably be a shortage of agricultural water. In fact, the study area is very vulnerable to drought, with drought occurring every 2–3 years and severe drought occurring every 7 years [14].

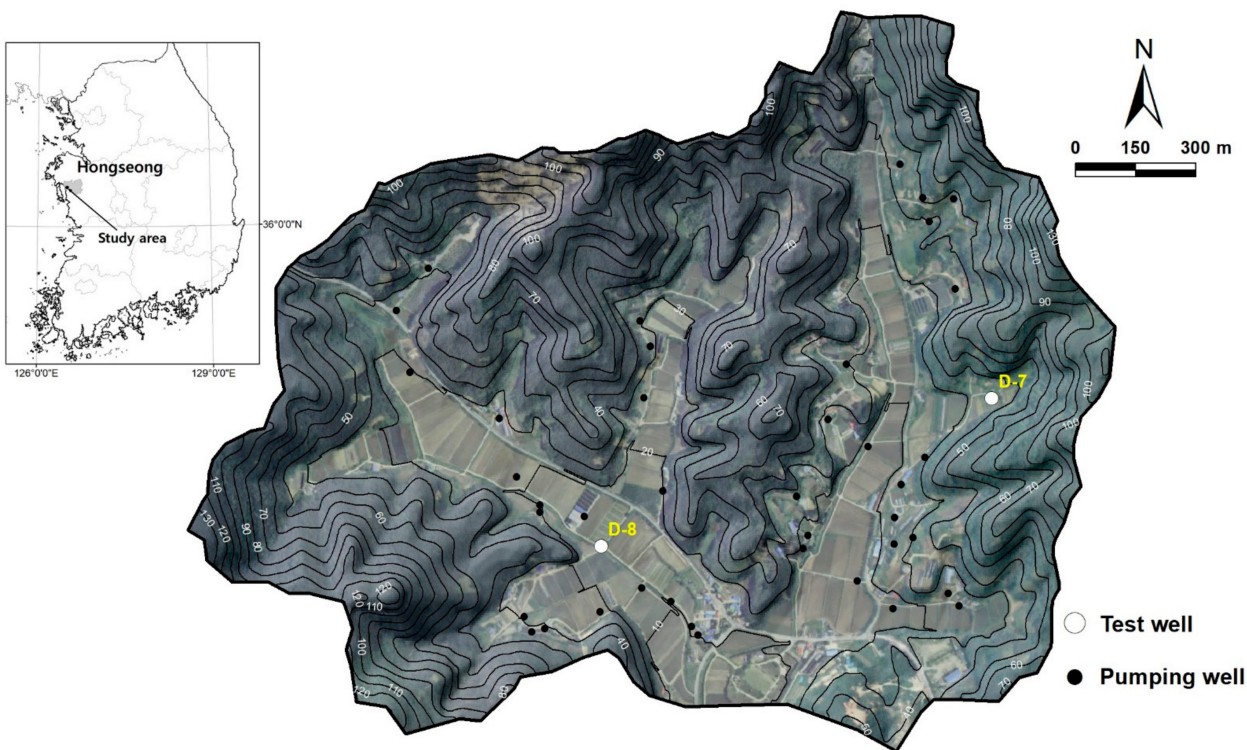

**Figure 1.** Study area and well locations.

## 2.2. Air Surging

Two wells were selected among the main pumping wells for well rehabilitation in the study area (Figure 1). The D-8 well was developed in 2012: the well depth is 103 m, and the well diameter is 350 mm. The admitted pumpage is 220 m$^3$/day and is used for agricultural purposes. Casing is installed to a depth of 60.5 m, and the screen sections are 24.5~28.0 m, 36.6~40.0 m, and 48.5~52.3 m in depth. The casing material is galvanized steel, and the pipe connected to the pump is made of PE (polyethylene). The section below 60.5 m was left in the bedrock without casing. The D-7 well was also developed in 2012: the well depth is 175 m, and the well diameter is 200 mm. The admitted pumpage is 90 m$^3$/day.

Casing is installed to a depth of 6 m, and the sections below 6 m in depth are designed to hold boreholes without casing in the bedrock. Both the D-7 and D-8 wells were developed for agricultural use (Table 1).

**Table 1.** Air surging wells for well rehabilitation.

| Well | D-8 | D-7 |
|---|---|---|
| Development (year) | 2012 | 2012 |
| Well depth (m) | 103 | 175 |
| Well diameter (mm) | 350 | 200 |
| Admitted pumpage ($m^3$/day) | 220 | 90 |
| Usage | Agricultural use | Agricultural use |
| Test date | 3–17 Novemer 2020 | 7–22 October 2021 |

Air surging was performed for the purpose of removing substances adsorbed on the casing and screen, as well as for suspended substances in the wells. Surging was conducted with pressurized air at about 2000 psi (150 atm), which was injected into the well to cover the casing, screen, and bedrock sections. The substances adsorbed to the casing and bedrock wall were removed using high air pressure, and the operation was performed sufficiently until the water coming out of the well was clear.

*2.3. Step Drawdown Test and Interpretation*

To compare and analyze the surging effects in each well, step drawdown tests were performed before and after surging. The step drawdown test is used for a comparative analysis of well performance or well yield by surging. In addition, the water flowing out of the well was sampled, and the substances contained in the sampled water were analyzed. The tests were conducted for the D-8 well from 3 to 7 November 2020, and for the D-7 well from 7 to 22 October 2021. The step drawdown test was carried out in 5 steps, with 1 h for each step. The pumping rate was adjusted to the maximum pumping rate in the last 5 steps based on the admitted pumpage of each well. The pumping rates of the D-8 well varied from 70.0 $m^3$/day to 165.2 $m^3$/day, and from 13.2 $m^3$/day to 81.9 $m^3$/day for the D-7 well.

The step drawdown test data were analyzed based on the Theis confined aquifer assumption [16,17]. The test wells were developed in fractured bedrock aquifers, and the upper alluvial layer or the less permeable layer was assumed to be a confining layer of the aquifer. This assumption is supported by the fact that there was little groundwater response at the monitoring well in the upper layer while pumping from the test well was in progress. The air surging effect on well performance or well yield was investigated by comparing the specific discharge, transmissivity, and drawdowns during a specific time period of 1 day at the pumping rate of each step.

The step drawdown test analysis in this study was based on the Birsoy and Summers' method [18]. According to Birsoy and Summers [18], an analytical solution for the drawdown response in a confined aquifer that is pumped at variable discharge rates can be expressed as the following Equation (1), applying the principle of superposition to Jacob's approximation of the Theis equation [17]:

$$s_n = \frac{2.3 Q_n}{4\pi T} \log\left\{ \left(\frac{2.25 T}{r^2 S}\right) \beta_{t(n)}(t - t_n) \right\} \tag{1}$$

where $\beta_{t(n)}(t - t_n)$ is the adjusted time as shown in Equation (2):

$$\beta_{t(n)}(t - t_n) = \prod_{i=1}^{n}(t - t_i)^{\Delta Q_i / Q_n} \tag{2}$$

$t_i$: time at which the *i*-th pumping period ended

$t - t_i$: time since the $i$-th pumping period started
$Q_i$: pumping rate during the $i$-th pumping period
$\Delta Q_i = Q_i - Q_n$: pumping rate increment beginning at time $t_i$.
The specific drawdown is converted into the following Equation (3) from Equation (1):

$$\frac{s_n}{Q_n} = \frac{2.3}{4\pi T} \log\left\{ \left(\frac{2.25T}{r^2 S}\right) \beta_{t(n)}(t - t_n) \right\} \tag{3}$$

In Equation (3), when $s_n/Q_n$ on the $y$-axis and $\log\left\{\beta_{t(n)}(t - t_n)\right\}$ on the $x$-axis are plotted and fitted, the transmissivity can be calculated through the slope ($\Delta$) of the fitted lines from Equation (4):

$$T = \frac{2.3}{4\pi\Delta} \tag{4}$$

If the slope is very different at each step, this indicates that the hydrogeologic boundaries are met or that there are some changes, such as inhomogeneity of the aquifer as pumping proceeds. In this case, it is practically meaningless to calculate the transmissivity of the aquifer [18]. In addition, if the fitted line is extended to a period of 1 day, the drawdown and specific discharge can be calculated when the pumping is continued to the time at the pumping rate of each step.

Well performance can be evaluated using the well efficiency before and after surging. Many methods for the evaluation of well efficiency have been developed and reported in the literature [19–27], but in this study, well efficiency was evaluated using Jacob's [22] and Rorabaugh's model [26]. Well efficiency is the ratio between the theoretical drawdown of the aquifer to the actual drawdown of the well [16,17]. The drawdown in a pumped well consists of two components, which are the aquifer losses and the well losses. A well performance test was conducted to determine these losses. Well efficiency, $E_w$, can be evaluated using the following Equation (5). Here, B, C, and p are the well parameters that were obtained from the step drawdown test:

$$E_w = \frac{BQ}{BQ + CQ^2} \text{ or, } E_w = \frac{BQ}{BQ + CQ^p} \tag{5}$$

The term BQ, which is the aquifer loss, represents the head losses caused by laminar flow in the aquifer and is proportional to the discharge. The terms, $CQ^2$ and $CQ^p$, which are the well loss, are non-linear terms, and represent turbulent flow in and around the well [16,17,22,25,27].

## 2.4. Analysis of Groundwater Quality

In order to evaluate the effect of improving groundwater quality by well surging, groundwater samples were collected during the pumping and surging of the D-8 and D-7 wells. Then, the collected groundwater samples were analyzed for major cations ($Ca^{2+}$, $Mg^{2+}$, $Na^+$, $K^+$, Fe, Mn, $Sr^{2+}$, and $SiO_2$), anions ($F^-$, $Cl^-$, $NO_3^-$, $NO_2^-$, and $SO_4^{2-}$), and dissolved organic carbon (DOC) using an inductively coupled plasma-optical emission spectrometer (Optima 7300DV ICP-OES, PerkinElmer Inc., Waltham, MA, USA), ion chromatography (ICS-1500, Dionex, Salt Lake City, UT, USA), and a total organic carbon analyzer (TOC-L, Shimazu Co., Tokyo, Japan), respectively. The spectroscopic properties of dissolved organic matter (DOM) were measured using a fluorometer (Aqualog, Horiba, Tokyo, Japan). Alkalinity was measured using the Gran titration method to determine the bicarbonate ($HCO_3^-$) concentration. The field parameters for temperature (T), pH, dissolved oxygen (DO), and electrical conductivity (EC) were continuously monitored during the pumping tests before and after surging.

## 2.5. Analysis of Clogging Substances

XRD, XRF, and SEM analyses were performed to reveal the identity of the clogging materials, which are substances deposited in wells or attached to borehole walls in the

bedrock. The substance samples were prepared using filtration through a 0.2 μm membrane filter from the turbid water that overflowed during surging. X-ray diffraction (XRD) and X-ray fluorescence (XRF) analyses were performed to reveal the minerals and chemical composition of these substances.

In the process of replacing well materials such as pumps and pipes after completing surging of the D-8 well, the substances deposited inside the pipes were recovered and analyzed using scanning electron microscopy (SEM) and energy dispersive X-ray spectroscopy (EDS) with the following procedures. The completely dried sample was adhered to an aluminum stub with double-sided adhesive carbon tapes and then coated with carbon (C) using an ion-sputter (208HR, Cressington Scientific Instruments Ltd., Watford, UK). For the observation device, a field emission scanning electron microscope (Quanta 650F, Thermo Fisher Scientific, Waltham, MA, USA) was used. Sample analysis was performed using a dual EDS system (XFlash 5010 SDD detector, Bruker Nano BmbH, Berlin, Germany) with an accelerating voltage of 20 kV. In addition, the compositions of the clogging substances from the pipe elements were analyzed using an inductively coupled plasma-optical emission spectrometer (Optima 7300DV ICP-OES, PerkinElmer Inc., Waltham, MA, USA).

## 3. Results and Discussion

### 3.1. Analysis of the Effects of Surging on Well Improvement

Figure 2 shows the air surging in the D-8 and D-7 wells. During the surging, brown colored turbid groundwater came out. The pictures taken by the borehole camera showed that the water quality significantly improved after surging. It is certain that the substances attached to the screens and casing of the D-8 well came off. The groundwater of the D-7 well was clean enough to observe the rock surface of the borehole after surging.

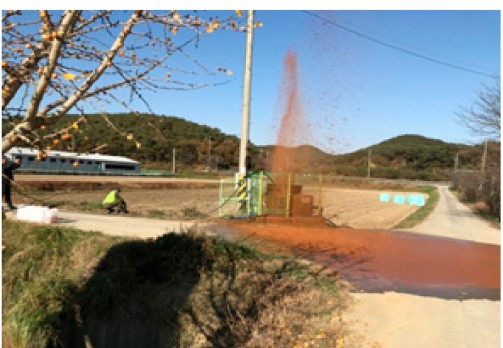 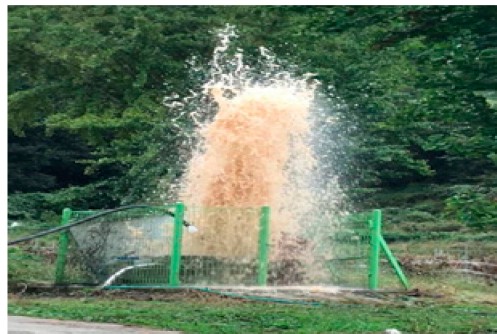

**Figure 2.** *Cont.*

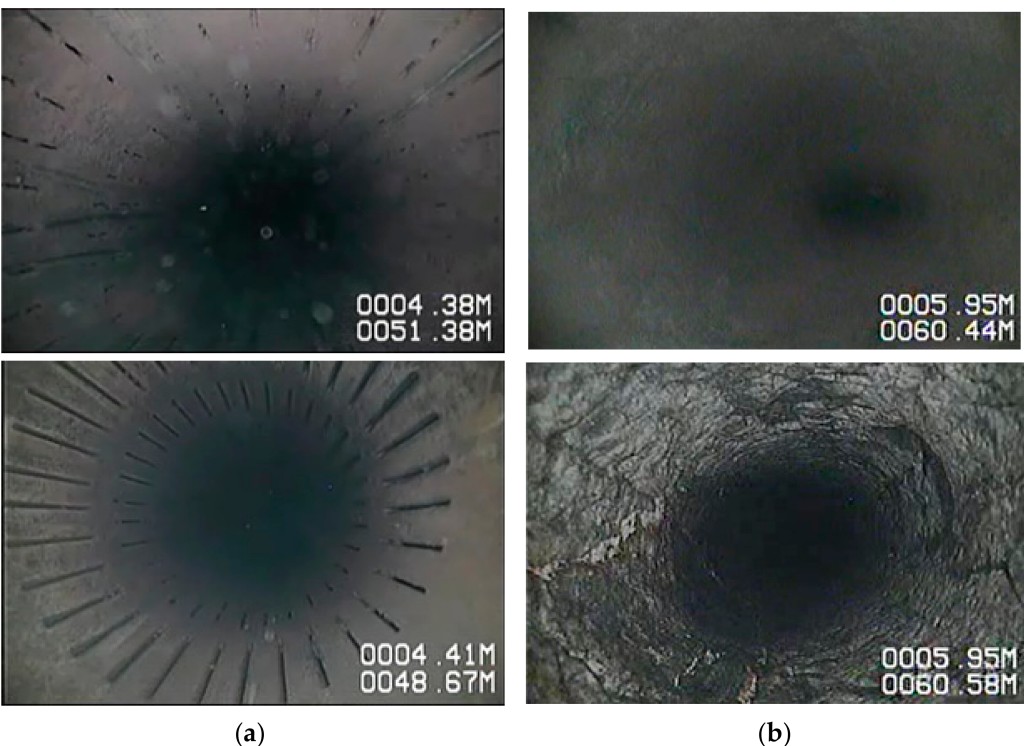

(**a**)                       (**b**)

**Figure 2.** Air surging operation (**top**) and well screen and groundwater quality comparisons before (**middle**) and after (**bottom**) surging from a borehole camera in the (**a**) D-8, and (**b**) D-7 wells.

Figure 3 shows the results of the step drawdown test before and after surging in the D-8 and D-7 wells. At each step, the drawdown decreased more after surging than before surging, so the productivity of the well was improved. In particular, the drawdowns of the D-7 well were greatly reduced.

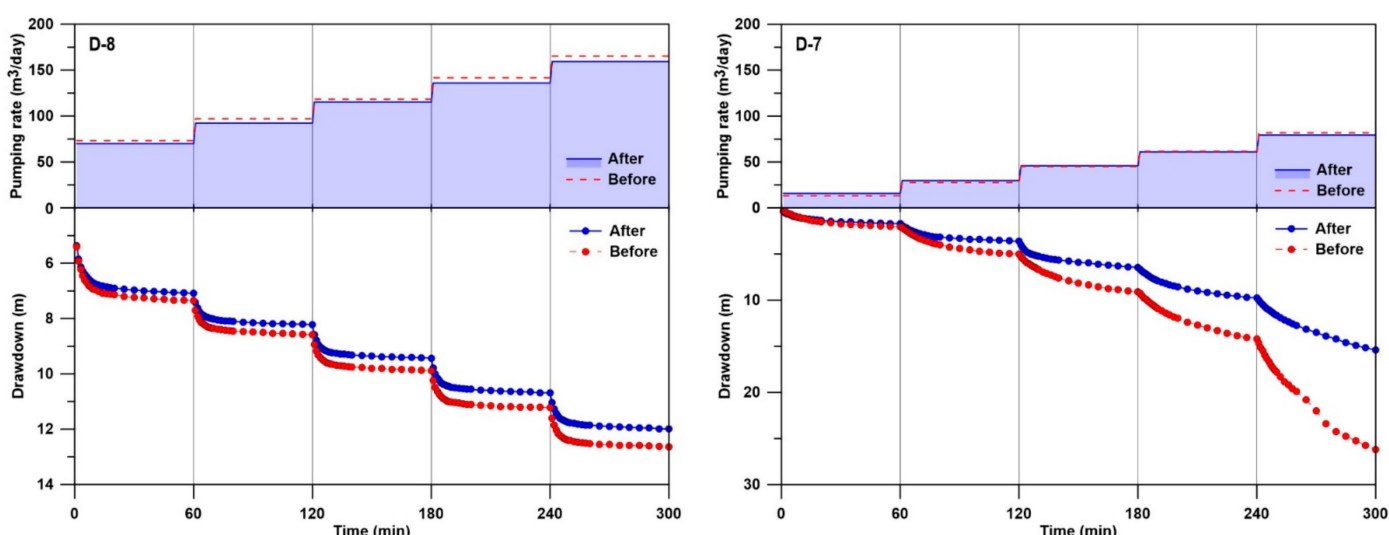

**Figure 3.** Groundwater drawdowns of the D-7 and D-8 wells during the step drawdown test.

In the step drawdown test, pumping was continuously performed. The productivity of a well needs to be compared on a specific time basis. Drawdowns, specific discharges, and transmissivities for the well productivities before and after surging were evaluated for 1 day. The drawdowns for 1 day at the pumping rate of each step were estimated by extrapolation of the fitting line between the log (adjusted time) and the specific drawdown

(s/Q) [17]. Additionally, transmissivities can be calculated using the slope of the fitted line for each step.

Figure 4 shows the fitted lines of the specific drawdown and log (adjusted time). In the initial phase of each step, the pumping rate fluctuated due to the wellbore storage and rapid drawdown, so curve fittings were conducted based on the latter part of the data. In an ideal confined aquifer, the slopes for each step should be similar, but in the D-7 well, these slopes are different, so well performance could not be evaluated by Jacob's or Rorabaugh's method [18]. Therefore, in this study, only the drawdown and specific discharge before and after surging and the slopes of the fitted lines at each step were compared.

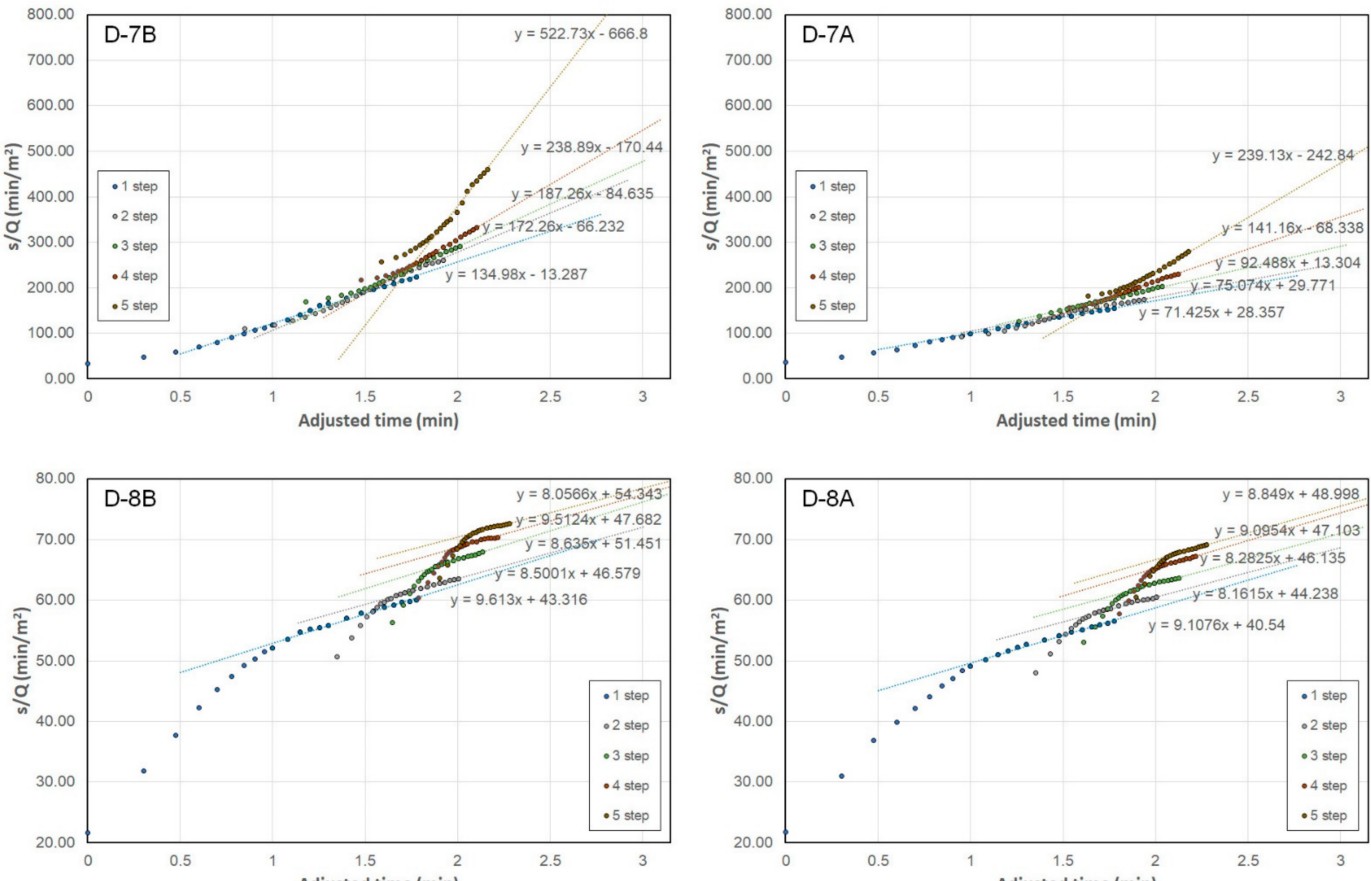

**Figure 4.** The relationship between specific drawdown and pumping rate before (D-8B, D-7B) and after (D-8A, D-7A) air surging. Curve fittings were made using the latter data for each pumping step.

Table 2 shows the quantitative analysis results of the step drawdown test before and after surging. In the step drawdown test before surging in the D-8 well, the pumping rates were from 73.2 m³/day in the first step to 165.2 m³/day in the fifth step; the drawdowns calculated on a daily basis were 3.37~9.15 m; the specific discharges were 18.05~19.61 m³/day/m; and the transmissivities were 27.43~32.73 m²/day. After surging, the calculated drawdowns on a daily basis were 3.37~8.52 m; the specific discharges were 18.71~20.78 m³/day/m; and the transmissivities were 28.95~32.31 m²/day at the pumping rates of 70.0 m³/day in the first step and 159.4 m³/day in the fifth step. It was seen that the average specific discharge before and after surging slightly increased from 18.80 m³/day/m to 19.79 m³/day/m, and the averagee transmissivity increased from 29.89 m²/day to 30.38 m²/day.

**Table 2.** Comparisons of specific discharges and transmissivities before and after air surging during each step in the D-8 and D-7 wells.

| Step | Before Surging | | | | After Surging | | | |
|------|------|------|------|------|------|------|------|------|
| | Q | $s_1$ | $Q/s_1$ | T | Q | $s_1$ | $Q/s_1$ | T |
| | (m³/day) | (m) | (m³/day/m) | (m²/day) | (m³/day) | (m) | (m³/day/m) | (m²/day) |
| (a) D-8 well | | | | | | | | |
| 1 | 73.2 | 3.75 | 19.54 | 27.43 | 70.0 | 3.37 | 20.78 | 28.95 |
| 2 | 97 | 4.95 | 19.61 | 31.02 | 92.4 | 4.49 | 20.57 | 32.31 |
| 3 | 118.4 | 6.39 | 18.53 | 27.72 | 115.2 | 5.78 | 19.92 | 31.84 |
| 4 | 141.6 | 7.74 | 18.29 | 30.54 | 136.0 | 7.16 | 18.99 | 28.99 |
| 5 | 165.2 | 9.15 | 18.05 | 32.73 | 159.4 | 8.52 | 18.71 | 29.80 |
| | Average | | 18.80 | 29.89 | Average | | 19.79 | 30.38 |
| (b) D-7 well | | | | | | | | |
| 1 | 13.2 | 3.79 | 3.49 | 1.95 | 15.9 | 2.80 | 5.67 | 3.69 |
| 2 | 27.7 | 9.19 | 3.01 | 1.53 | 29.8 | 5.52 | 5.40 | 3.51 |
| 3 | 45.2 | 15.91 | 2.84 | 1.41 | 45.9 | 9.74 | 4.71 | 2.85 |
| 4 | 61.7 | 25.03 | 2.47 | 1.10 | 61.1 | 16.02 | 3.81 | 1.87 |
| 5 | 81.9 | 55.97 | 1.46 | 0.50 | 79.4 | 28.25 | 2.81 | 1.10 |
| | Average | | 2.65 | 1.30 | Average | | 4.48 | 2.61 |

Q: pumping rate, $s_1$: the estimated drawdown after one day pumping, $Q/s_1$: specific discharge, T: transmissitivity.

In the D-7 well, the pumping rates were from 13.2 m³/day in the first step to 81.9 m³/day in the fifth step before surging; the drawdowns calculated on a daily basis were 3.79~55.97 m; the specific discharges were 1.46~3.49 m³/day/m; and the transmissivities were 0.50~1.95 m²/day. After surging, drawdowns at the pumping rates from 15.9 m³/day in the first step to 79.4 m³/day in the fifth step; the calculated drawdowns on a daily basis were 2.80~28.25 m; the specific discharges were 2.81~5.67 m³/day/m; the transmissivities were 1.10~3.69 m²/day. The average specific discharge before and after surging increased significantly from 2.65 m³/day/m to 4.48 m³/day/m, and the average transmissivity increased from 1.30 m²/day to 2.61 m²/day.

The specific discharge and transmissivity in the D-7 well decreased as pumping progressed from the first to the fifth step. This indicates that less permeable hydrogeologic boundaries were encountered or that the aquifer properties were changed by the inhomogeneity of the aquifer. However, since the purpose of this study was to evaluate well improvement before and after surging, these phenomena should be elucidated through additional research.

Figure 5 shows the relationship between the drawdown and pumping rate calculated on a daily basis for each step. The drawdowns after surging were clearly reduced compared with before surging. The surging effects on well improvement were clearly shown based on the drawdown and the adjusted time even if the field conditions did not satisfy the ideal confined aquifer assumptions. According to Jacob's equation, a linear relationship appears in the relationship between the pumping rate and drawdown [16]. A linear relationship was clearly observed in the D-8 well, but the drawdowns in the D-7 well were much larger than the linear increase in drawdown according to the pumping rate, and the best fitting was achieved with an exponential function. The exponential relationship between drawdown and pumping rate may be due to the effect of the hydrogeologic boundary or aquifer inhomogeneity, as mentioned above. The ideal confined aquifer model seems to fit better in the bedrock aquifer of the D-8 well than the D-7 well.

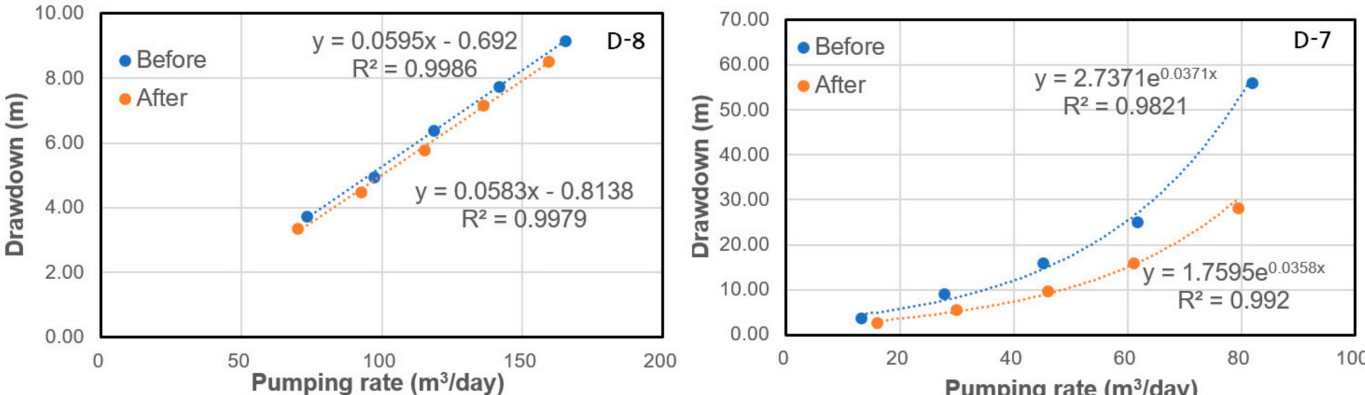

**Figure 5.** The relationship between the estimated one-day drawdown and pumping rate before and after surging in the D-8 and D-7 wells.

From the fitting equation of the D-8 well, for example, at a pumping rate of 50 m$^3$/day and 200 m$^3$/day, the drawdowns for 1 day were estimated to be 2.28 m and 11.21 m before surging, and 2.10 m and 10.85 m after surging, respectively. In the D-7 well, this effect was even greater, and at a pumping rate of 20 m$^3$/day and 80 m$^3$/day, the drawdowns for 1 day were expected to be reduced from 5.75 m and 53.24 m before surging to 3.60 m and 30.85 m after surging, respectively.

To evaluate the well performance, well efficiency was calculated from the step draw-down test [16,17]. Table 3 shows the results of calculating the well efficiency by the Jacob's [22] and Rorabaugh [27] equations. The Rorabaugh equation calculated the well parameters using Labadie–Helweg's method [26]. The well efficiencies of the D-8 well before surging were estimated to be 68.5~83.1% and 40.8~51.7% according to the Jacob method and Labadie–Helweg's method, respectively. After surging, they were estimated to be 67.3~82.4% according to the Jacob method and 41.3~52.3% according to Labadie–Helweg's method. There was a slight improvement effect in the D-8 well even though the well efficiency did not show much change (Table 3). In the D-7 well, the well efficiencies before surging were estimated to be 52.4~87.2%, 50.2~77.4%, and 48.9~82.7%, 54.7~81.3% after surging according to Jacob's and Labadie–Helweg's methods, respectively (Table 3).

**Table 3.** Comparisons of well parameters (B, C, P) estimated by Jacob's and Labadie–Helweg's methods in the (a) D-8 and (b) D-7 wells before and after air surging.

| | | | | | | | | |
|---|---|---|---|---|---|---|---|---|
| **(a) D-8 Well** | | | | | | | | |
| Before surging | | | | | | | | |
| Step | Q (m$^3$/day) | $s_w$ (m) | Jacob's method (B = $3.40 \times 10^{-2}$, C = $9.72 \times 10^{-5}$) | | | Labadie–Helweg's method (B = $2.10 \times 10^{-2}$, C = $1.93 \times 10^{-3}$, P = 1.54) | | |
| | | | BQ | CQ$^2$ | W.E. (%) | BQ | CQ$^P$ | W.E. (%) |
| 1 | 73.2 | 3.05 | 2.55 | 0.52 | 83.1 | 1.54 | 1.44 | 51.7 |
| 2 | 97 | 4.28 | 3.38 | 0.91 | 78.7 | 2.04 | 2.21 | 47.9 |
| 3 | 118 | 5.59 | 4.13 | 1.36 | 75.2 | 2.49 | 3.01 | 45.3 |
| 4 | 142 | 6.92 | 4.94 | 1.95 | 71.7 | 2.98 | 3.96 | 42.9 |
| 5 | 165 | 8.33 | 5.76 | 2.65 | 68.5 | 3.47 | 5.03 | 40.8 |

**Table 3.** *Cont.*

| | | | | **(a) D-8 Well** | | | | |
|---|---|---|---|---|---|---|---|---|
| | | | | After surging | | | | |
| Step | Q (m³/day) | $s_w$ (m) | Jacob's method (B = 3.26 × 10⁻², C = 9.93 × 10⁻⁵) | | | Labadie–Helweg's method (B = 2.10 × 10⁻², C = 1.84 × 10⁻³, P = 1.54) | | |
| | | | BQ | CQ² | W.E. (%) | BQ | CQ^P | W.E. (%) |
| 1 | 70.0 | 2.75 | 2.29 | 0.49 | 82.4 | 1.40 | 1.28 | 52.3 |
| 2 | 92.4 | 3.88 | 3.02 | 0.85 | 78.1 | 1.85 | 1.96 | 48.5 |
| 3 | 115.2 | 5.09 | 3.76 | 1.32 | 74.0 | 2.30 | 2.75 | 45.6 |
| 4 | 136.0 | 6.35 | 4.44 | 1.84 | 70.7 | 2.72 | 3.55 | 43.4 |
| 5 | 159.4 | 7.65 | 5.20 | 2.52 | 67.3 | 3.19 | 4.54 | 41.3 |
| | | | | **(b) D-7 Well** | | | | |
| | | | | Before surging | | | | |
| Step | Q (m³/day) | $s_w$ (m) | Jacob's method (B = 0.14, C = 1.50 × 10⁻³) | | | Labadie–Helweg's method (B = 0.12, C = 6.50 × 10⁻³, P = 1.67) | | |
| | | | BQ | CQ² | W.E. (%) | BQ | CQ^P | W.E. (%) |
| 1 | 13.2 | 2.05 | 1.80 | 0.26 | 87.2 | 1.65 | 0.48 | 77.4 |
| 2 | 27.7 | 5.02 | 3.78 | 1.16 | 76.5 | 3.46 | 1.66 | 67.6 |
| 3 | 45.2 | 9.13 | 6.17 | 3.09 | 66.6 | 5.64 | 3.75 | 60.1 |
| 4 | 61.7 | 14.23 | 8.42 | 5.75 | 59.4 | 7.70 | 6.31 | 55.0 |
| * 5 | 81.9 | 26.21 | 11.18 | 10.13 | 52.4 | 10.22 | 10.13 | 50.2 |
| | | | | After surging | | | | |
| Step | Q (m³/day) | $s_w$ (m) | Jacob's method (B = 8.80 × 10⁻², C = 1.16 × 10⁻³) | | | Labadie–Helweg's method (B = 9.33 × 10⁻², C = 2.38 × 10⁻³, P = 1.80) | | |
| | | | BQ | CQ² | W.E. (%) | BQ | CQ^P | W.E. (%) |
| 1 | 15.9 | 1.71 | 1.40 | 0.29 | 82.7 | 1.48 | 0.34 | 81.3 |
| 2 | 29.8 | 3.61 | 2.62 | 1.03 | 71.9 | 2.78 | 1.06 | 72.5 |
| 3 | 45.9 | 6.44 | 4.04 | 2.44 | 62.4 | 4.28 | 2.29 | 65.1 |
| 4 | 61.1 | 9.75 | 5.38 | 4.32 | 55.5 | 5.70 | 3.83 | 59.8 |
| * 5 | 79.4 | 15.39 | 6.99 | 7.29 | 48.9 | 7.41 | 6.14 | 54.7 |

Q: pumping rate, $s_w$: drawdown, W.E.: well efficiency (BQ/(BQ + CQ²) or BQ/(BQ + CQ^P)). * Step 5 data were not used in the fitting, and the results presented in step 5 were derived from the results of steps 1 to 4.

In the D-7 well, At the D-7 well, a significant change in drawdown in the step drawdown test occurred, but the corresponding improvement in the well efficiency was negligible. This is probably because the model applied to calculate the well efficiency of the D-7 well did not fit well with the ideal confined aquifer assumptions according to Equation (1). In particular, the fifth-step data were not related enough to the previous step data to fit the Jacob's or Rorabaugh's model, so the fifth-step data were excluded from the fitting. In Table 3b, the well efficiency results of the fifth step were derived from the results of step 1 to 4. However, in the D-8 well, the fitting between the specific drawdown and the pumping rate was properly made, and it was judged that the confined aquifer assumption was relatively well-satisfied. In the D-7 well, the well parameters changed with time while meeting the inhomogeneity and hydrogeologic boundary of the aquifer during the pumping. In such a situation, it seems difficult to evaluate the surging effect using well efficiency. Instead, it is necessary to comprehensively evaluate the drawdown, specific discharge, and transmissivity based on a certain time period. The surging effect on well improvement

could be evaluated by considering both the aquifer parameters and the well parameters at the same time for the D-8 well, but, due to the heterogeneity of the bedrock aquifer and the hydrogeologic boundary effects of the D-7 well, it is difficult to evaluate the surging effect using only the well parameters.

*3.2. Groundwater Quality Changes before and after Air Surging*

The effect of improving the quality of the groundwater was investigated by comparing the groundwater quality changes during the pumping tests before and after surging. Table 4 shows the analytical results of the major ions, DOC, and fluorescence indices of the DOM for groundwater samples collected from the pumping and surging tests. The fluorescence peaks and indices of dissolved organic matter are described in detail in the literature [28–30].

Figure 6a shows the monitoring results of the field groundwater quality parameters of electrical conductivity (EC), pH, and dissolved oxygen (DO) during two pumping tests (PP-I and PP-II) before and after surging. The field water quality factors of the D-8 and D-7 wells were gradually stabilized after 1~2 h in both of the pumping tests, except for the DO of D-7 (Figure 6a). During all the pumping tests before and after surging, the EC values of D-7 were higher than those of D-8, but the pH values of D-7 were lower than those of D-8. In the two wells (D-8 and D-7) located in the east and west of the study area, the EC and pH according to the pumping before and after surging showed different patterns. The EC values indicating the total amount of dissolved components were higher in the pumping test after surging in the D-7 well than before surging, but the opposite patterns were shown in the D-8 well. However, the pH value showed the opposite trend to that of EC. The increase in EC and the decrease in pH during the pumping test of the D-7 well after surging seem to be related to the inflow of groundwater containing contaminants in the shallow strata close to the surface. This inflow of shallow groundwater was also confirmed by the increased dissolved oxygen (0.33~3.14 mg/L) in the late pumping test (PP-II) compared to the low dissolved oxygen (0.08~0.23 mg/L) in the pumping test (PP-I) before surging. Figure 6b shows the changes in the major cations ($Ca^{2+}$, $Na^+$, and $SiO_2$), anions ($HCO_3^-$, $Cl^-$, $NO_3^-$, and $SO_4^{2-}$), total dissolved solids (TDS), dissolved organic carbon (DOC), and groundwater quality index (GWQI) observed during surging and two pumping tests (the groundwater quality of the D-7 well was not measured in PP-I). GWQI was used as an indicator of the overall characteristics and changes in groundwater quality, and in this study, it was calculated using the method by Adimalla and Qian [31]. Most of the dissolved components measured in the D-8 well were rather high in the surging process, which is thought to be due to the dissolution in the surging process of the materials deposited or coated inside the well. In addition, it was confirmed that $Na^+$, $Cl^-$, $NO_3^-$, and $SO_4^{2-}$ of the D-8 well were slightly more increased in the second pumping stage (PP-II) than in the first pumping stage (PP-I), although the difference was not large. The increase in these components seems to be due to the effect of pollutants (such as nutrient salts and chemical fertilizers) supplied from agricultural land around the D-8 well. Unlike the D-8 well, D-7 showed almost no difference in water quality between the surging stage and the second stage of pumping (PP-II). The different characteristics of the D-8 and D-7 wells for surging are thought to be due to the differences in the inflow of shallow groundwater according to the pumping of wells, as was confirmed from the field water quality factors.

**Table 4.** The analysis of results from dissolved ions and dissolved organic matter in groundwater collected during surging and pumping tests.

| Sample | | TDS | Major Dissolved Ions | | | | | | | | | | | | | | | CBE | DOC | FDOM | | | | | | |
|---|---|---|---|---|---|---|---|---|---|---|---|---|---|---|---|---|---|---|---|---|---|---|---|---|---|---|
| | | | Ca | Mg | Na | K | Fe | Mn | SiO$_2$ | Sr | HCO$_3$ | F | Cl | NO$_2$ | NO$_3$ | SO$_4$ | | | Peak C | Peak A | Peak M | Peak T | HIX | BIX | FI |
| | | mg/L | mg/L | mg/L | mg/L | mg/L | mg/L | mg/L | mg/L | mg/L | mg/L | mg/L | mg/L | mg/L | mg/L | mg/L | (%) | mg/L | QSU | QSU | QSU | QSU | | | |
| **(a) D-8** | | | | | | | | | | | | | | | | | | | | | | | | | | |
| Pumping Phase I | BP0 | 127.2 | 21.8 | 8.0 | 9.5 | 1.21 | <0.1 | <0.1 | 18.9 | 0.10 | 103.7 | 0.19 | 12.0 | 0.19 | 2.18 | 2.37 | 1.4 | 0.379 | 0.008 | 0.013 | 0.008 | 0.007 | 0.685 | 1.062 | 1.602 |
| | BP1 | 139.6 | 22.2 | 7.9 | 9.4 | 1.24 | <0.1 | <0.1 | 18.8 | 0.10 | 100.4 | 0.13 | 22.5 | 0.55 | 3.57 | 4.37 | −5.1 | 0.345 | 0.008 | 0.013 | 0.010 | 0.004 | 0.737 | 1.083 | 2.631 |
| | BP2 | 129.9 | 26.6 | 8.2 | 9.7 | 1.22 | 0.12 | <0.1 | 18.5 | 0.13 | 113.6 | 0.07 | 7.3 | 0.10 | 1.22 | 1.05 | 7.4 | 0.333 | 0.009 | 0.011 | 0.010 | 0.007 | 0.741 | 0.992 | 1.675 |
| | BP3 | 140.3 | 27.2 | 8.2 | 9.8 | 1.24 | 0.20 | <0.1 | 18.4 | 0.13 | 117.6 | 0.00 | 12.6 | 0.18 | 2.22 | 2.45 | 2.5 | 0.373 | 0.010 | 0.014 | 0.008 | 0.005 | 0.780 | 0.942 | 2.256 |
| | BP4 | 142.9 | 27.7 | 8.4 | 9.9 | 1.29 | 0.37 | 0.05 | 18.5 | 0.14 | 121.1 | 0.00 | 12.7 | 0.19 | 2.19 | 2.40 | 2.4 | 0.323 | 0.010 | 0.013 | 0.012 | 0.013 | 0.706 | 1.167 | 2.445 |
| | BP5 | 143.8 | 27.7 | 8.5 | 10.0 | 1.26 | 0.49 | 0.06 | 19.0 | 0.14 | 119.1 | 0.00 | 13.4 | 0.19 | 2.28 | 2.51 | 3.0 | 0.345 | 0.010 | 0.012 | 0.009 | 0.006 | 0.766 | 0.778 | 1.614 |
| Surging Stage | SU1 | 131.8 | 22.2 | 8.1 | 10.7 | 1.30 | <0.1 | 0.05 | 16.5 | 0.10 | 86.4 | 0.13 | 22.5 | 0.55 | 3.57 | 4.37 | 1.4 | 0.501 | 0.022 | 0.033 | 0.032 | 0.073 | 0.655 | 1.329 | 1.773 |
| | SU2 | 115.7 | 18.5 | 6.8 | 10.3 | 1.37 | <0.1 | <0.1 | 14.0 | 0.08 | 69.2 | 0.13 | 24.5 | 0.54 | 1.17 | 4.71 | 0.6 | 0.842 | 0.058 | 0.084 | 0.060 | 0.080 | 0.809 | 0.935 | 1.691 |
| | SU3 | 160.2 | 30.9 | 8.2 | 11.0 | 1.76 | <0.1 | <0.1 | 16.1 | 0.14 | 127.8 | 0.17 | 23.2 | 0.49 | 0.00 | 5.91 | −2.5 | 0.620 | 0.044 | 0.072 | 0.045 | 0.058 | 0.799 | 0.840 | 1.705 |
| | SU4 | 167.3 | 30.8 | 8.3 | 10.9 | 1.70 | <0.1 | <0.1 | 16.8 | 0.15 | 133.6 | 0.14 | 23.8 | 0.30 | 2.15 | 6.78 | −5.3 | 0.422 | 0.031 | 0.051 | 0.041 | 0.083 | 0.727 | 1.066 | 1.770 |
| | SU5 | 171.2 | 32.5 | 8.6 | 11.3 | 1.77 | <0.1 | 0.05 | 17.7 | 0.16 | 133.0 | 0.15 | 27.0 | 0.73 | 0.57 | 6.25 | −3.8 | 0.397 | 0.024 | 0.041 | 0.034 | 0.064 | 0.717 | 1.056 | 1.669 |
| | SU6 | 166.2 | 31.6 | 8.4 | 11.2 | 1.62 | <0.1 | <0.1 | 17.3 | 0.15 | 129.1 | 0.14 | 24.3 | 0.61 | 1.55 | 6.50 | −3.1 | 0.408 | 0.031 | 0.048 | 0.024 | 0.010 | 0.876 | 0.746 | 1.596 |
| Pumping Phase II | AP0 | 129.3 | 21.6 | 7.6 | 10.1 | 1.22 | <0.1 | 0.05 | 17.9 | 0.10 | 80.3 | 0.15 | 24.9 | 0.80 | 0.77 | 5.56 | 0.5 | 0.406 | 0.010 | 0.018 | 0.010 | 0.008 | 0.750 | 0.792 | 1.642 |
| | AP1 | 145.0 | 24.9 | 8.1 | 10.3 | 1.40 | <0.1 | 0.05 | 18.4 | 0.12 | 104.3 | 0.15 | 22.3 | 0.69 | 3.79 | 4.38 | −2.2 | 0.387 | 0.012 | 0.015 | 0.011 | 0.006 | 0.732 | 0.809 | 1.809 |
| | AP2 | 147.5 | 25.6 | 8.0 | 10.2 | 1.40 | 0.26 | 0.07 | 18.0 | 0.12 | 107.3 | 0.13 | 22.8 | 0.72 | 3.74 | 4.52 | −2.7 | 0.367 | 0.012 | 0.017 | 0.012 | 0.008 | 0.729 | 0.798 | 2.069 |
| | AP3 | 147.7 | 26.5 | 8.3 | 10.9 | 1.44 | 0.37 | 0.08 | 18.5 | 0.13 | 100.3 | 0.13 | 23.6 | 0.74 | 3.86 | 4.63 | 1.1 | 0.367 | 0.011 | 0.018 | 0.014 | 0.009 | 0.689 | 0.897 | 1.877 |
| | AP4 | 156.5 | 27.2 | 8.4 | 10.9 | 1.43 | 0.51 | 0.09 | 18.3 | 0.13 | 113.0 | 0.13 | 25.2 | 0.73 | 4.07 | 4.83 | −3.1 | 0.415 | 0.013 | 0.015 | 0.012 | 0.006 | 0.779 | 0.785 | 1.742 |
| | AP5 | 157.6 | 27.3 | 8.5 | 11.1 | 1.42 | 0.68 | 0.10 | 18.3 | 0.13 | 114.1 | 0.13 | 25.1 | 0.68 | 4.06 | 4.76 | −2.7 | 0.395 | 0.012 | 0.016 | 0.012 | 0.007 | 0.775 | 0.900 | 1.954 |
| **(b) D-7** | | | | | | | | | | | | | | | | | | | | | | | | | | |
| Surging Stage | SU1 | 134.1 | 22.3 | 7.5 | 11.3 | 1.58 | <0.1 | <0.1 | 14.8 | 1.20 | 105.1 | 0.22 | 9.3 | 0.83 | 6.0 | 9.1 | −0.3 | 0.741 | 0.017 | 0.021 | 0.015 | 0.018 | 0.653 | 0.774 | 1.461 |
| | SU2 | 135.5 | 23.8 | 7.2 | 11.3 | 1.01 | <0.1 | <0.1 | 15.3 | 1.01 | 103.1 | 0.19 | 9.9 | 0.83 | 8.3 | 7.6 | 0.7 | 0.551 | 0.014 | 0.017 | 0.011 | 0.022 | 0.586 | 0.942 | 1.698 |
| | SU3 | 133.8 | 24.2 | 7.2 | 10.0 | 0.96 | <0.1 | <0.1 | 15.1 | 0.81 | 99.2 | 0.18 | 10.3 | 0.75 | 9.6 | 7.3 | 0.8 | 0.576 | 0.011 | 0.014 | 0.009 | 0.013 | 0.613 | 0.983 | 1.692 |
| | SU4 | 134.1 | 24.7 | 7.3 | 9.5 | 0.95 | <0.1 | <0.1 | 15.1 | 0.75 | 97.8 | 0.17 | 10.6 | 0.78 | 10.6 | 7.0 | 0.9 | 0.456 | 0.008 | 0.010 | 0.008 | 0.010 | 0.551 | 0.866 | 1.794 |
| | SU5 | 135.9 | 24.7 | 7.4 | 9.5 | 0.95 | <0.1 | <0.1 | 15.2 | 0.76 | 101.5 | 0.17 | 10.6 | 0.82 | 10.2 | 7.1 | 0.0 | 0.650 | 0.008 | 0.012 | 0.008 | 0.013 | 0.547 | 0.924 | 1.208 |
| Pumping Phase II | AP1 | 134.7 | 21.7 | 7.5 | 11.8 | 1.10 | <0.1 | <0.1 | 15.7 | 1.40 | 109.6 | 0.34 | 8.8 | 0.72 | 4.9 | 8.8 | −1.8 | 0.605 | 0.011 | 0.006 | 0.004 | 0.008 | 0.693 | 0.724 | 1.808 |
| | AP2 | 132.9 | 22.6 | 7.0 | 10.7 | 1.02 | <0.1 | <0.1 | 15.5 | 1.12 | 105.0 | 0.34 | 9.4 | 0.70 | 6.7 | 7.8 | −1.8 | 0.484 | 0.014 | 0.011 | 0.007 | 0.009 | 0.669 | 0.882 | 1.868 |
| | AP3 | 132.2 | 23.9 | 6.8 | 10.4 | 1.00 | <0.1 | <0.1 | 15.8 | 0.75 | 96.9 | 0.19 | 10.4 | 0.69 | 9.6 | 6.4 | 1.1 | 0.514 | 0.012 | 0.007 | 0.007 | 0.011 | 0.607 | 0.721 | 1.536 |
| | AP4 | 131.4 | 24.6 | 6.5 | 9.2 | 0.93 | <0.1 | <0.1 | 15.8 | 0.58 | 94.3 | 0.18 | 10.7 | 0.65 | 11.4 | 5.7 | 0.6 | 0.577 | 0.010 | 0.007 | 0.006 | 0.011 | 0.643 | 0.688 | 2.247 |
| | AP5 | 132.5 | 25.2 | 6.4 | 8.8 | 0.89 | <0.1 | <0.1 | 15.8 | 0.52 | 94.1 | 0.17 | 10.9 | 0.66 | 12.4 | 5.7 | 0.2 | 0.522 | 0.008 | 0.005 | 0.006 | 0.009 | 0.650 | 0.696 | 1.598 |

In groundwater wells in areas with active agricultural activities, biological activities such as biofilm formation actively occur due to the increase in organic matter flowing into the groundwater, and the clogging of wells can be easily observed. Graham et al. [32] reported research results regarding increases in the elution of dissolved organic carbon and dissolved organic matter from the biofilm and soil organic matter on the aquifer porous media into the groundwater by pumping. In this study, an analysis of dissolved organic carbon and fluorescent dissolved organic matter was performed in order to investigate the removal effect of such organic matter according to the surging. Figure 7 shows the results of the 3D-EEM (3-dimensional excitation emission matrix) of groundwater fluorescent dissolved organic matter (FDOM) collected during well surging and the two pumping tests. Similar to the previous results for the major ion components (Figure 7), higher concentrations of FDOM were observed in the surging stage than in all the pumping tests in the D-8 well. In addition, although the difference was not very large, it was found that the concentration of FDOM showed a somewhat greater increase in the second pumping phase (PP-II) than in the first pumping phase (PP-I). However, in the case of the D-7 well, the difference in FDOM between the surging stage and the second pumping phase (PP-II) was not large and similar to the results of the major ion components. The excitation (Ex) and emission (Em) wavelength values of the representative components of the FDOM, as shown in Figure 7, were as follows: Peak A (Ex260/Em450), Peak C (Ex275/Em440), Peak M (Ex300/Em390), and Peak T (Ex275/Em340) [30]. The FDOM observed in the groundwater collected during well surging of D-8 mainly consisted of humic-like (increase in peak A and peak C) and protein-like (increase in peak T) substances; anthropogenic pollutants (peak M) from agricultural activities were also observed [29]. The fluorescence index (FI) is an index that identifies the relative contribution of dissolved organic matter from terrestrial and microbial sources. FI from 1.2 to 1.5 indicate that humic-like materials derived from microorganisms is dominant, and that from 1.7 to 2.0 indicate that humic-like materials of terrestrial origin are dominant [33]. During the surging and pumping of the D-8 and D-7 wells, the FI of some groundwater samples was lower than 1.7, but most of the groundwater showed a value of 1.7 or higher, indicating humic-like substances of terrestrial origin.

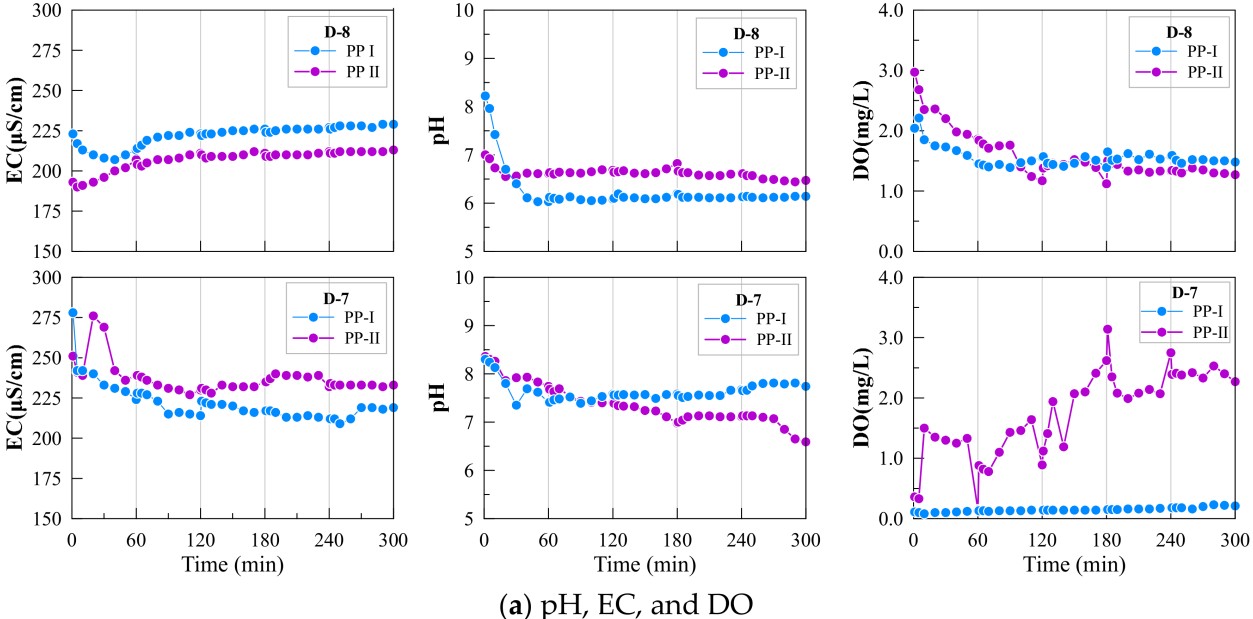

(**a**) pH, EC, and DO

**Figure 6.** *Cont.*

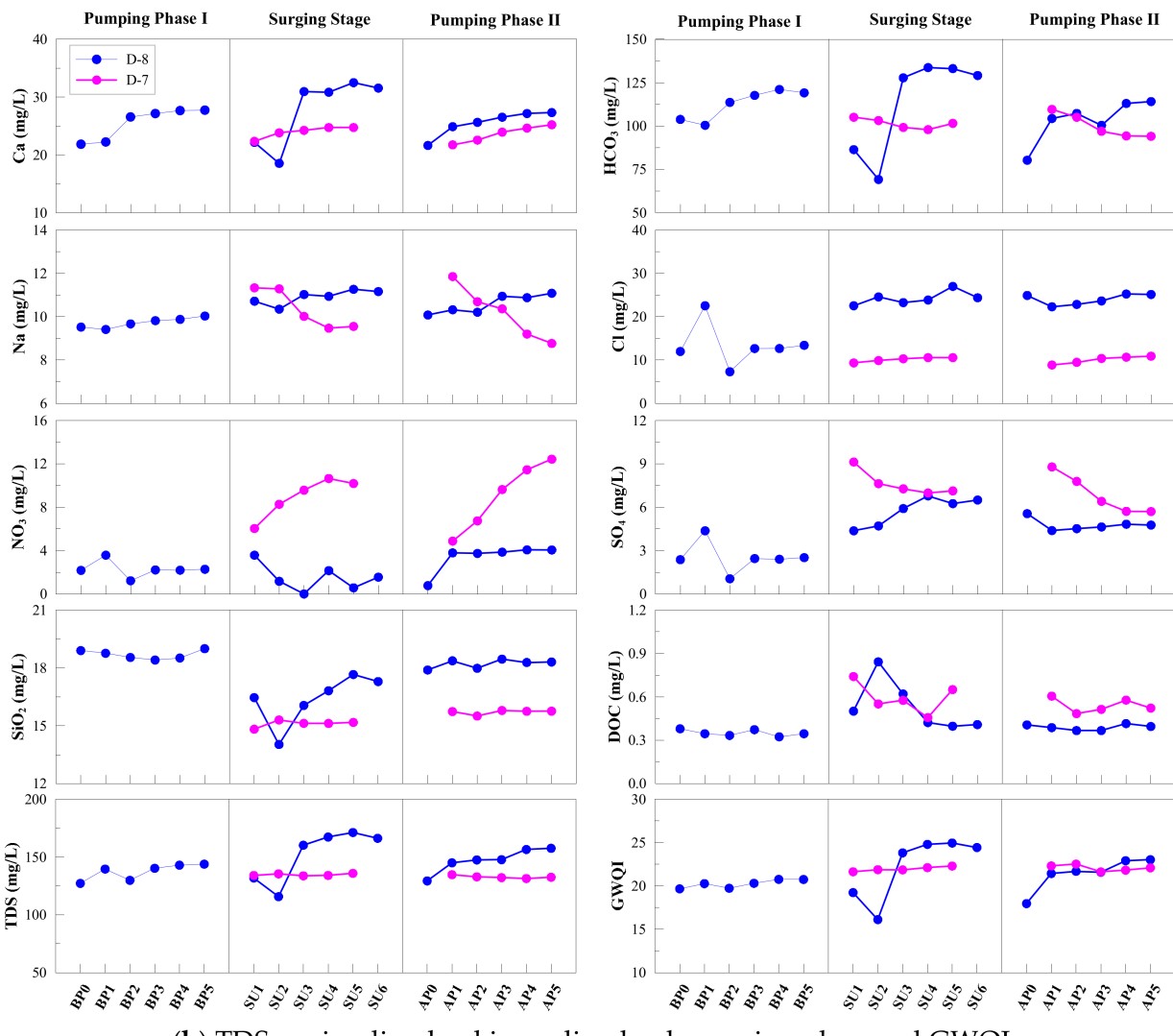

(**b**) TDS, major dissolved ions, dissolved organic carbon and GWQI

**Figure 6.** The results of (**a**) changes in field groundwater quality parameters observed during pumping tests and (**b**) variation in the of major ions, total dissolved solids (TDS), dissolved organic carbon, and groundwater quality index (GWQI) in groundwater samples collected during pumping and surging in the D-8 and D-7 wells.

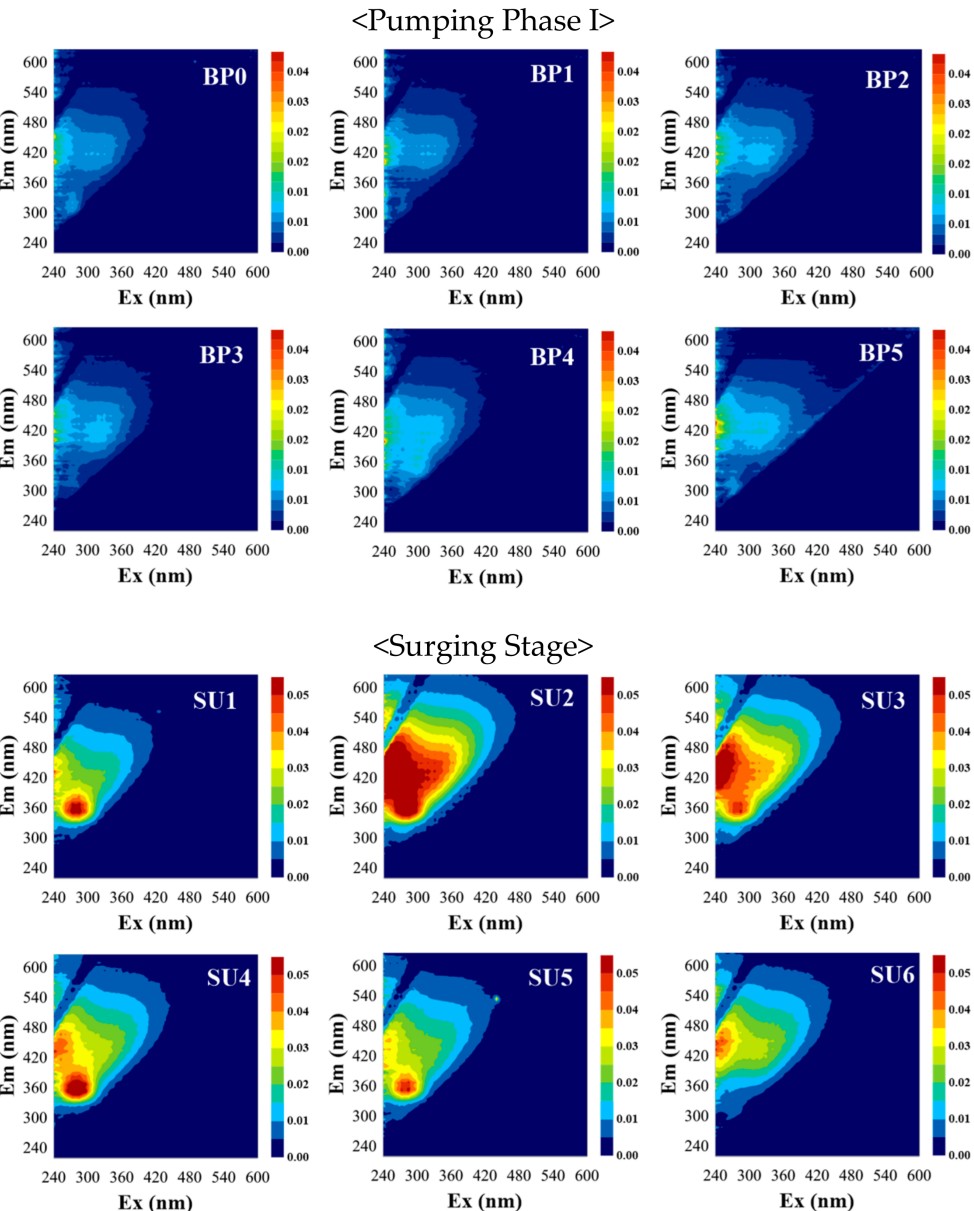

**Figure 7.** *Cont.*

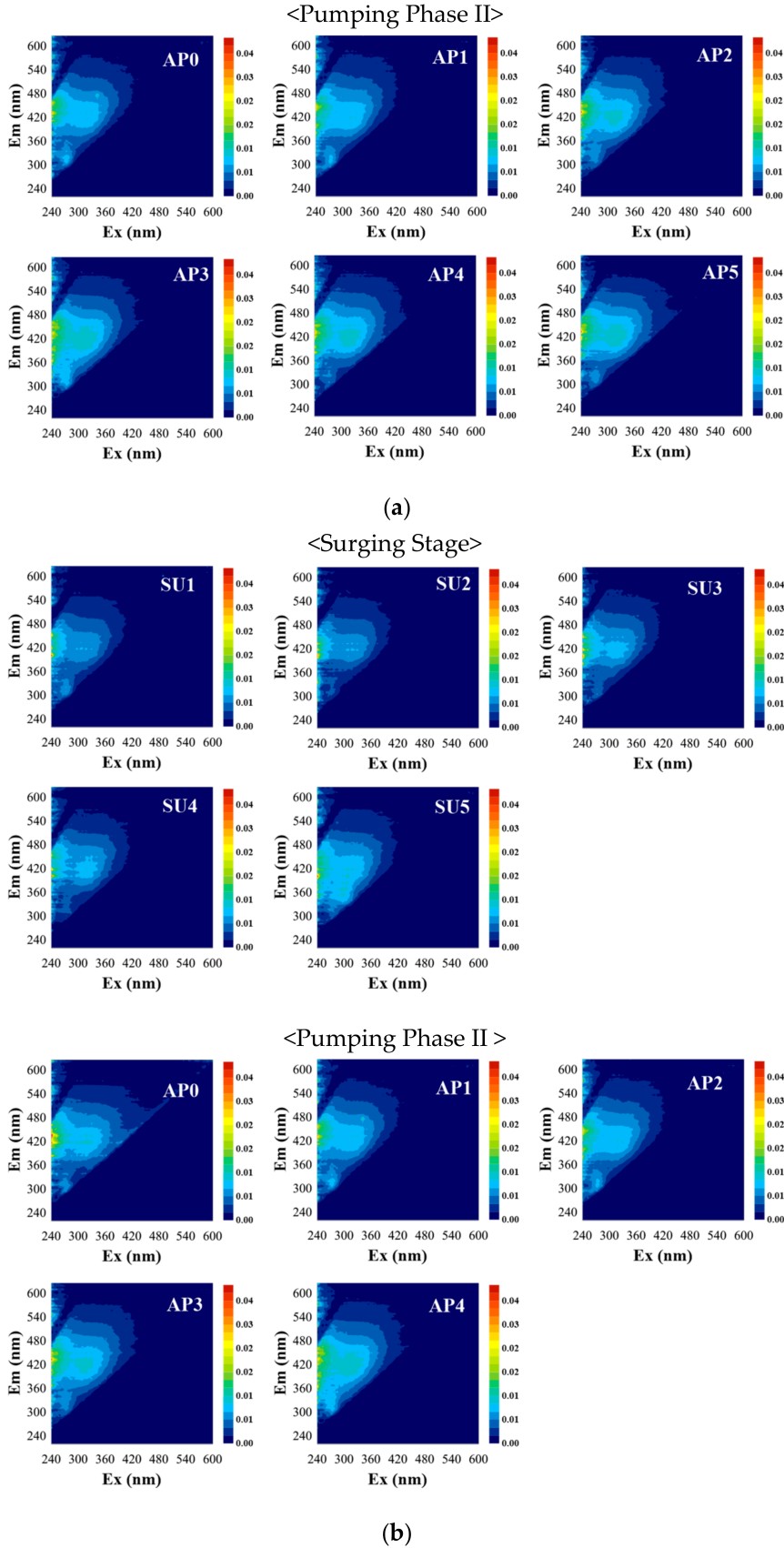

(**a**)

(**b**)

**Figure 7.** Variation in fluorescence EEM off dissolved organic matter (DOM) of groundwater during pumping tests performed before and after well surging in the (**a**) D-8 and (**b**) D-7 well.

### 3.3. Analysis of Substances Blocking the Well Screen

Table 5 shows the results of XRD analysis of the substances flowing out with the groundwater from the D-8 well during the surging process. Surging was carried out until the turbid water became clear, and was finished within 2 h in both the D-8 and D-7 wells. During the surging processes, the turbid waters were collected from the early to late stages in order: six samples from the D-8 well and five samples from the D-7 well. As the results of the analyses show, most of the substances that made overflowing water turbid were amorphous solids or non-crystalline solids (Figure 8). Among them, the crystalline solids analyzed in the XRD analysis are shown in Table 5.

**Table 5.** XRD analysis results of the substances mixed with the overflowing water during the air surging process.

| Mineral | Chemical Formula | Surging Stage in the D-8 Well Early → Late | | | | | | Surging Stage in the D-7 Well Early → Late | | | | |
|---|---|---|---|---|---|---|---|---|---|---|---|---|
| | | D-8(1) | D-8(2) | D-8(3) | D-8(4) | D-8(5) | D-8(6) | D-7(1) | D-7(2) | D-7(3) | D-7(4) | D-7(5) |
| Quartz | $SiO_2$ | ■ | | ■ | ■ | ■ | ■ | ■ | ■ | ■ | ■ | ■ |
| Illite-2M1 | $(K,H_3O)AlSi_3AlO_{10}(OH)_2$ | | ■ | ■ | ■ | ■ | ■ | ■ | ■ | ■ | ■ | ■ |
| Kaolinite-1A | $Al_2Si_2O_5(OH)_4$ | | | ■ | ■ | ■ | ■ | ■ | ■ | ■ | ■ | ■ |
| Muscovite | $KAl_2 (Si_3Al)O_{10}(OH)_2$ | | ■ | ■ | ■ | ■ | ■ | ■ | ■ | ■ | ■ | ■ |
| Montmorillonite-15A | $CaO_2(Al,Mg)_2Si_4O_{10}(OH)_214H_2O$ | | | | | | | ■ | ■ | ■ | ■ | ■ |
| Chlorite | $Mg_2Al_3 (Si_3Al)O_{10}(O)_8$ | | | | | | | ■ | ■ | ■ | ■ | ■ |
| Jacobsite, syn | $MnFe_2O_4$ | ■ | ■ | ■ | ■ | ■ | ■ | | | | | |
| Lepidocrocite | $FeO(OH)$ | | ■ | ■ | ■ | ■ | ■ | | | | | |
| Albite, ordered | $NaAlSi_3O_8$ | | | | ■ | | ■ | ■ | ■ | | | |
| Albite, calcian | $(Na,Ca)(Si,Al)_4O_8$ | | | | | | | | | ■ | ■ | ■ |
| Hematite | $Fe_2O_3$ | | | | | | | | | | ■ | ■ |
| Orthoclase | $KAlSi_3O_8$ | | | | | | | ■ | | | | |

The numbers in ( ) indicate the sampling order of overflowing water from the well during air surging. The symbol ■ indicates detection in the analysis.

During the early stage of surging in the D-8 well, quartz and jacobsite appeared, and during the late stages, muscovite, albite, and lepidocrocite, including illite and kaolinite as clay minerals, were identified. In the D-7 well, quartz, muscovite, and clay minerals such as illite, kaolinite, montmorillonite, and chlorite were included in the flowing-out substances from the early to later stages of the surging. However, these solid materials were mostly amorphous, and there were few minerals with a definite crystalline structure.

Figure 9a shows a cut-off section of the pipe connected to the pump. Brown-colored materials several millimeters thick piled up on the wall of the pipe. Figure 9b shows the results of the SEM observation of the materials. According to the results of the SEM-EDS analysis for the determination of what elements these materials have, iron and oxygen were the main constituents, and in terms of weight percentage, iron (Fe) accounted for more than 50%. These results indicate that Fe-related substances accounted for most of the deposit materials.

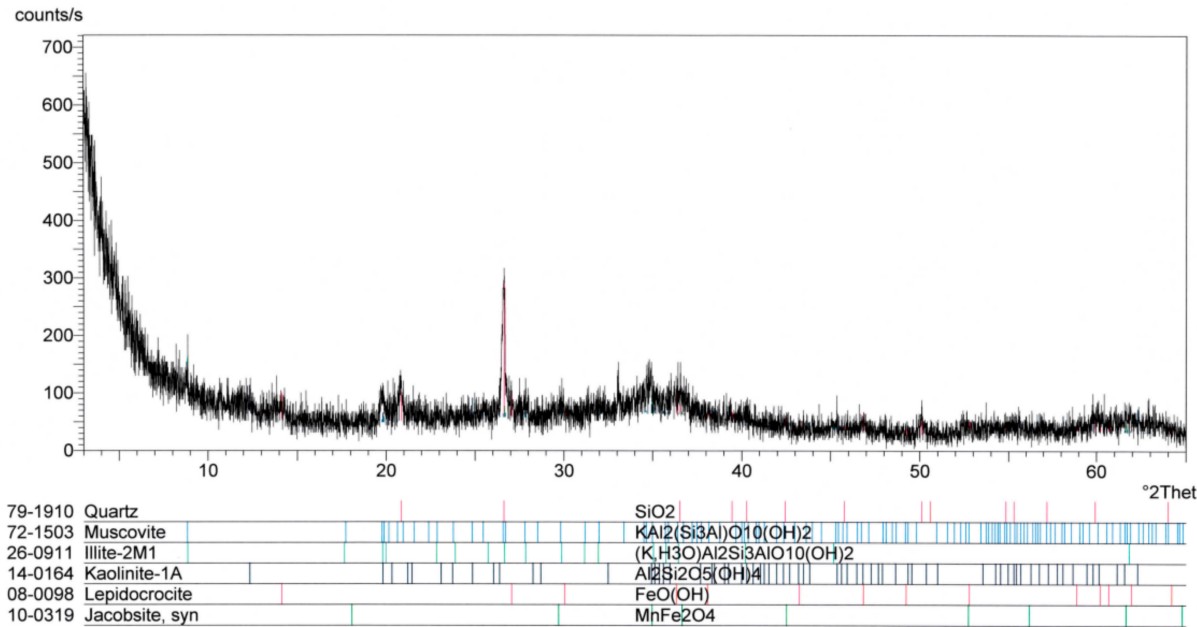

| 79-1910 | Quartz | SiO2 |
| 72-1503 | Muscovite | KAl2(Si3Al)O10(OH)2 |
| 26-0911 | Illite-2M1 | (K,H3O)Al2Si3AlO10(OH)2 |
| 14-0164 | Kaolinite-1A | Al2Si2O5(OH)4 |
| 08-0098 | Lepidocrocite | FeO(OH) |
| 10-0319 | Jacobsite, syn | MnFe2O4 |

**Figure 8.** An example of the XRD analysis results of the substances collected during surging (for the D-8 well (3)). Some crystalline solids or minerals are as shown in the figure, but the other minerals except for quartz are not clear enough to distinguish peaks, and the substances are mostly amorphous.

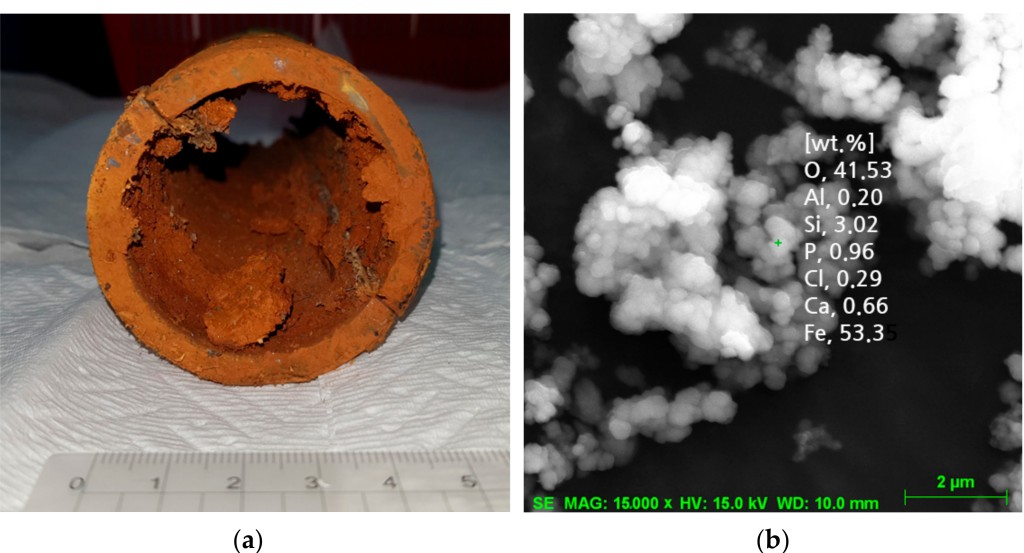

(**a**)                                                   (**b**)

**Figure 9.** SEM-EDS analysis result for the material deposited inside the pipe of the D-8 well, (**a**) Brown-colored substances inside the pipe, and (**b**) SEM electron image for the deposit, which is a typical, representative picture of the particles that make up the sample. The green + is the point where chemical analysis was performed.

Table 6 shows the results of the analysis of the elements using ICP-OES for the material deposited in the pipe of the D-8 well. The materials collected from the upper and lower parts were analyzed, and the compositions of each material were similar. According to the SEM-EDS analysis results, Fe accounted for most of the weight percent with 45.4~47.3%. For other elements, arsenic (As), zinc (Zn), and vanadium (V) accounted for a large portion. Arsenic (As) often coexists with Fe in nature, and zinc (Zn) is an element that forms the well casing. Therefore, it can be assumed that iron oxide in the aquifer dissolves together with arsenic into groundwater over time. Additionally, it is possible that Fe and Zn were eluted as the well casing was corroded.

**Table 6.** Chemical analysis results of the materials deposited inside the pipe.

(a) Major compositions

|  |  |  |  |  |  |  |  |  | (Unit: wt.%) |
|---|---|---|---|---|---|---|---|---|---|
| Sample | Na | Mg | Al | K | Ca | Mn | Fe | Ti | Others |
| D-8_U | 0.02 | 0.03 | 0.03 | 0.01 | 0.49 | 0.20 | 45.4 | ND |  |
| D-8_L | 0.01 | 0.03 | 0.02 | ND | 0.50 | 0.05 | 47.3 | ND |  |

(b) Others

|  |  |  |  |  |  |  |  |  |  |  |  | (Unit: mg/kg) |
|---|---|---|---|---|---|---|---|---|---|---|---|---|
| Sample | Cu | Zn | Sr | Cd | Li | Cr | Co | Ni | V | As | Mo | Pb |
| D-8_U | 28 | 487 | 66 | 2.0 | ND | 62 | 19 | 27 | 198 | 542 | 5 | 7 |
| D-8_L | 4 | 398 | 69 | 1.9 | ND | 64 | 4 | 27 | 237 | 687 | 4 | 2 |

(a) ND: not detected (<MDL), MDL: method detection limit (Ti: 0.0025 wt.%, K: 0.0050 wt.%). (b) ND: not detected (<MDL), method detection limit = 1.00 mg/kg. D-8-U and D-8_L refer to the upper and lower sections, respectively, of the sediment deposited inside the pipe.

## 4. Conclusions

In this study, air surging, the most commonly used physical method among well rehabilitation techniques, was performed, and its improvements were evaluated in bedrock aquifers. In two wells developed in a bedrock aquifer for agricultural uses, step draw-down tests were conducted, and the water qualities and substances deposited inside the wells were analyzed. To analyze the step drawdown test data, the Birsoy and Summers' method [17], under the Theis assumptions of the ideal confined aquifer, was applied. The changes in drawdowns, specific discharges, and transmissivities of each step based on a reference time (in this study, 1 day according to the adjusted time) were compared. Draw-downs were reduced in both the D-8 and D-7 test wells. Accordingly, the average specific discharges were increased by 5.3% and 68.8% and the average transmissivities by 1.6% and 100.4%, respectively. However, well efficiency could not be evaluated using the Jacob or Rorabaugh models for the D-7 well due to factors such as the uncertainty of the aquifer model, aquifer inhomogeneity, and the hydrogeologic boundary. Since well efficiency has many factors requiring consideration in order to evaluate well performance, it seems better to compare the pumping rate and drawdown based on the reference time calculated by the adjusted time in practice.

The improvement in groundwater quality was investigated by analyzing the ground-water quality during the pumping tests before and after surging. The increase in EC and decrease in pH during the pumping test of the D-7 well after surging seem to be related to the inflow of groundwater containing contaminants in the shallow strata close to the surface. Most of the dissolved components of the D-8 well were present at high levels during the surging process, and the FDOM observed in the groundwater collected during the well surging of D-8, mainly consisting of humic-like and protein-like substances; anthropogenic pollutants from agricultural activities were also observed. The groundwater quality of the D-7 well showed almost no difference between the surging and pumping stages. The fluorescence index (FI) for the groundwater of the D-8 and D-7 wells mostly showed a value of 1.7 or higher, indicating humic-like substances of terrestrial origin.

The materials collected during surging and the substances deposited inside the well pipe were analyzed using XRD, XRF, and SEM-EDS. The materials were mostly amorphous, and there were few minerals with a definite crystalline structure. According to the SEM-EDS analysis, Fe-related substances accounted for most the deposit materials, such as Fe oxides. Fe accounted for the highest percentage of the weightof the materials deposited inside the pipes of the D-8 well, with 45.4~47.3%. The other elements, arsenic and zinc, were found to exist together with iron or could be eluted from the well casing.

In order to prevent the deterioration of well performance and water quality through the above results, attention should also be paid to casing material and screen design in

the early stage of well development. Additionally, for the maintenance of the well, it is necessary to periodically check the quantity and water quality, monitor the inflow of pollutants from the surface, and take appropriate measures for well rehabilitation and pollution prevention.

**Author Contributions:** Conceptualization, methodology, software, validation, formal analysis, investigation, writing—original draft preparation, writing—review and editing, and supervision, K.H. and K.-S.K.; resources, data preparation, H.A., E.L., S.L. and H.C.K. All the authors significantly contributed to manuscript preparation. All authors have read and agreed to the published version of the manuscript.

**Funding:** This research was funded by the Korea Environment Industry and Technology Institute (KEITI) through the Demand Responsive Water Supply Service Program (or Project), funded by the Korean Ministry of Environment (MOE)(146526).

**Institutional Review Board Statement:** Not applicable.

**Informed Consent Statement:** Not applicable.

**Data Availability Statement:** Data is available on request.

**Conflicts of Interest:** The authors declare no conflict of interest.

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
