# Peer review of "Evaluation of Well Improvement and Water Quality Change before and after Air Surging in Bedrock Aquifers"

_water, doi:10.3390/w14142233_

Round 1
Reviewer 1 Report
The manuscript is suitable for publishing in Water. It contains many experimental results. Before publishing some issues should be addressed:
1. Introduction: The authors should clearly present the aim of their research.
2. Results and Discussion: I miss a discussion of results as there are only results presented and compared with each other.
3. Conclusion: I miss a final evaluation of how the taken measures helped. I also miss some suggestions on how to prevent well deterioration and water quality decline in the future after the measures that were taken.
4. Figures and tables should be embedded into the text, as close to the reference as possible, according to Instructions for authors.
Author Response
Responses to the reviewers’ comments
Manuscript ID: water-1762216
Type of manuscript: Article
Title: Evaluation of well improvement and water quality change before and after air surging in bedrock aquifers
Authors: Kyoochul Ha, Hyowon An, Eunhee Lee, Sujeong Lee, Hyoung Chan Kim and Kyung-Seok Ko*
We have carefully considered all comments from three reviewers and provided point-to-point responses in the revised version. And the manuscript was proofread through the English editing service, and the certificate is presented as follows. Corrected English text is displayed in red in the manuscript.
Reviewer 1
The manuscript is suitable for publishing in Water. It contains many experimental results. Before publishing some issues should be addressed:
Re: Thank you for your valuable comments and suggestions. We have considered all comments and suggestions and revised our manuscript to justify our works.
- Introduction: The authors should clearly present the aim of their research.
Re: Thank you for your valuable comments and suggestions. The purpose of this study is presented from line 106 to line 110. However, as suggested by the reviewer, the following content was added from line 110 to line 115 in order to present the aim of the research more clearly.
“The improvement effects of air surging are mainly achieved by comparing the well efficiencies by the step drawdown tests. However, in bedrock aquifers, it is difficult to estimate well efficiency because the aquifer are not homogeneous and the hydrogeologic boundary conditions are varied. Therefore, this study presented a practical evaluation methods for step drawdown test. In addition, we intend to contribute to future well maintenance by providing information on substances that degrade well performance.”
- Results and Discussion: I miss a discussion of results as there are only results presented and compared with each other.
Re: Discussions are presented with each result in the section of the Results and Discussion. Each result was presented and further discussion was conducted with tables and figures. For example, from line 349 to line 353 the relationship between drawdown and pumping rate was discussed. In addition, for each analysis result, discussioin is inserted through an additional figure if necessary.
- Conclusion: I miss a final evaluation of how the taken measures helped. I also miss some suggestions on how to prevent well deterioration and water quality decline in the future after the measures that were taken.
Re: Thank you for your suggestions. As suggested by the reviewer, the following sentences were added to the manuscript to present what actions should be taken in the future based on the research results from line 567 to line 573.
In order to prevent deterioration of well performance and water quality through the above results, attention should also be paid to casing material and screen design in the early stage of well development. And, for the maintenance of the well, it is necessary to periodically check the quantity and water quality, monitor the inflow of pollutants from the surface, and take appropriate measures for well rehabilitation and pollution prevention.
- Figures and tables should be embedded into the text, as close to the reference as possible, according to Instructions for authors.
Re: Thank you for your suggestions. The manuscript was prepared in the format required by Water Journal. Figures, tables, and references have been reviewed once again and will be edited by the MDPI Water Editorial Office in the future.

Reviewer 2 Report
The paper requires revisions:
1-Authors used "the analysis results of XRD, XRF, and SEM-EDS for the substances collected during surging and the substances deposited inside the well pipe, they found that most of the substances were Fe-related amorphous components. " Authors need justifications for these results. Authors are recommended to improve results by using more robust statistical analysis.
2-Introuction can be improved by adding the paper entitled "Reliability evaluation of groundwater quality index using data-driven models".
3-Authors need to add a new section about field study in details
4-Authors can calculate WQI for the groundwater and present their evaluation for this case study.
Author Response
Responses to the reviewers’ comments
Manuscript ID: water-1762216
Type of manuscript: Article
Title: Evaluation of well improvement and water quality change before and after air surging in bedrock aquifers
Authors: Kyoochul Ha, Hyowon An, Eunhee Lee, Sujeong Lee, Hyoung Chan Kim and Kyung-Seok Ko*
We have carefully considered all comments from three reviewers and provided point-to-point responses in the revised version. And the manuscript was proofread through the English editing service, and the certificate is presented as follows. Corrected English text is displayed in red in the manuscript.
Reviewer 2
1-Authors used "the analysis results of XRD, XRF, and SEM-EDS for the substances collected during surging and the substances deposited inside the well pipe, they found that most of the substances were Fe-related amorphous components. " Authors need justifications for these results. Authors are recommended to improve results by using more robust statistical analysis.
Re: Figure 9(b) is a typical, representative SEM electron image of the particles constituting the sample. If you look at the chemical analysis results for the points marked in green, it can be seen that Fe accounts for the majority at 53.3 wt.%. In EDS, We only know the chemistry. And, it is possible to know the presence of crystals. It's hard to say that the particles in this picture are crystalline. EDS analysis results show that the material deposited in the pipe consists of iron oxides with Si and minor ions. In the XRF pattern of Fig 8, quartz is clearly present, but the rest of the minerals are not clear (The background effect due to amorphous), which is indicating that the materials have amorphous characteristics. In Figure 8, the sentence was modified to clarify that the materials are mostly amorphous as the following.
Some crystalline solids or minerals are as shown in the figure, but the other minerals except for quartz are not so clear to distinguish peaks, and the substances are mostly amorphous.
2-Introduction can be improved by adding the paper entitled "Reliability evaluation of groundwater quality index using data-driven models".
Re: Thank you for your suggestions. Although the data driven technique is somewhat far from the purpose of this study, water quality evaluation of existing wells need to be conducted for the utilization of the well during the droughts. The following sentence was added to the Introduction as suggested by the reviewer from line 51 to line 55. And, the mentioned reference was added (from line 585 to 586).
“In addition, it is worth considering a plan to utilize the existing wells through water quality evaluation using the recently developed data driven technique to determine whether the appropriate water quality is satisfied for each well.”
3-Authors need to add a new section about field study in details.
Re: The detailed process for field study is described in section of 2. Materials and Methods. And, Table 1 has information about the test wells, and Figures 1 and 2 show the location and surging scene.
4-Authors can calculate WQI for the groundwater and present their evaluation for this case study.
Re: According to the reviewer’s opinion, the authors calculated groundwater quality index (GWQI), which is presented in addition to Fig. 6. Changes in TDS during pumping are additionally presented in Fig. 6. And, the following sentences were modified and a reference was added to the manuscript (from line 429 to line 434, from line 641 to 643)
“total dissolved solids (TDS), dissolved organic carbon (DOC), and groundwater quality index (GWQI) observed during surging and two pumping tests (The groundwater quality of the D-7 well was not meas-ured in PP-I). GWQI has been used as an indicator of the overall charac-teristics and changes of groundwater quality, and in this study, it was calculated using method by Adimalla and Qian [31].”

Round 2
Reviewer 1 Report
I would like to thank the authors for considering the comments.
I think, the article has been improved, all my comments have been incorporated and the paper may be published.
Just one remark for the authors, that the purpose of the study is not the same as the aim of the study.
Reviewer 2 Report
Accept as is